# DREAMIX: VIDEO DIFFUSION MODELS ARE GENERAL VIDEO EDITORS

Input Video

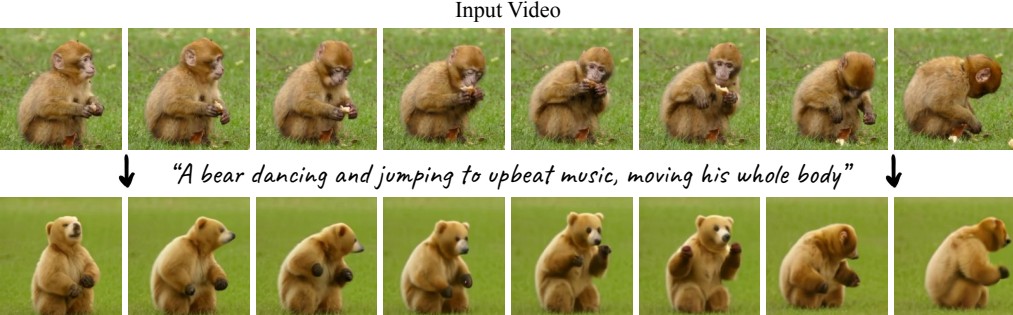

*"A bear dancing and jumping to upbeat music, moving his whole body"*

Figure 1: ***Video Editing with Dreamix***: By conditioning on the text prompt "A bear dancing and jumping to upbeat music, moving his whole body", Dreamix transforms the eating monkey (top row) into a dancing bear (bottom row), affecting motion and appearance. It maintains fidelity to color, object and camera pose, and results in a temporally consistent video. *We strongly encourage the reviewer to view the supplementary videos*

## ABSTRACT

Text-driven image and video diffusion models have recently achieved unprecedented generation realism. While diffusion models have been successfully applied for image editing, none can edit **motion** in video. We present the first diffusion-based method that is able to perform text-based motion and appearance editing of general, real-world videos. Our approach uses a video diffusion model to combine, at inference time, the low-resolution spatio-temporal information from the original video with new, high resolution information that it synthesized to align with the guiding text prompt. As maintaining high-fidelity to the original video requires retaining some of its high-resolution information, we add a preliminary stage of finetuning the model on the original video, significantly boosting fidelity. We propose to improve motion editability by using a mixed objective that jointly finetunes with full temporal attention and with temporal attention masking. We extend our method for animating images, bringing them to life by adding motion to existing or new objects, and camera movements. Extensive experiments showcase our method's remarkable ability to edit motion in videos.

## 1 INTRODUCTION

Recent advancements in generative models Ho et al. (2020); Chang et al. (2022); Yu et al. (2022a); Chang et al. (2023) and multimodal vision-language models Radford et al. (2021), paved the way to large-scale text-to-image models capable of unprecedented generation realism and diversity Ramesh et al. (2022); Rombach et al. (2022); Saharia et al. (2022b); Nichol et al. (2021); Avrahami et al. (2022b). These models have ushered in a new era of creativity, applications, and research. Although these models offer new creative processes, they are limited to synthesizing *new* images rather than editing *existing* ones. To bridge this gap, intuitive image editing methods offer text-based editing of generated and real images while maintaining some of their original attributes Hertz et al. (2022); Tumanyan et al. (2022); Brooks et al. (2022); Kawar et al. (2022); Valevski et al. (2022). Similarly to images, text-to-video models have recently been proposed Ho et al. (2022c;a); Singer et al. (2022); Yu et al. (2022b), but very few methods use them for video editing and none can edit the **motion** in videos.

In text-guided video editing, the user provides an input video and a text prompt describing the desired attributes of the resulting video (Fig. 1). The objectives are three-fold: i) alignment: the edited video should conform with the input text prompt ii) fidelity: the edited video should preserve the content of the original input iii) quality: the edited video should be of high-quality. Video editing is more challenging than image editing, as it requires synthesizing new **motion**, not merely modifying appearance. It also requires **temporal consistency**. As a result, applying image-level editing methods e.g. SDEdit Meng et al. (2021) or Prompt-to-Prompt Hertz et al. (2022) sequentially on the video frames is insufficient.

We present a new method, Dreamix, to adapt a text-conditioned video diffusion model (VDM) for video editing, in a manner inspired by UniTune Valevski et al. (2022). The core of our method is enabling a text-conditioned VDM to maintain high fidelity to an input video via two main ideas. First, instead of using pure-noise as initialization for the model, we use a degraded version of the original video, keeping only low spatio-temporal information by downscaling it and adding noise. This is similar to SDEdit but the degradation includes not merely noise, but also downscaling. Second, we further improve the fidelity to the original video by finetuning the model on the original video. Finetuning ensures the model has knowledge of the high-resolution attributes of the original video. Naively finetuning on the input video results in relatively low motion editability as the model learns to prefer the original motion instead of following the text prompt. We propose a novel use for the mixed finetuning approach, suggested in VDM Ho et al. (2022c) and Imagen-Video Ho et al. (2022a), in which the VDMs are trained on both images and video. In our approach, we finetune the model on both the original video but also on its (unordered) frames individually. This allows us to perform significantly larger motion edits with high fidelity to the original video.

As a further contribution, we leverage our video editing model to add motion to still images (see Fig. 2) e.g., animating the objects and background in an image or creating dynamic camera motion. To do so, we first create a coarse video by simple image processing operations, e.g., frame replication or geometric image transformation. We then edit it with our Dreamix video editor. Our framework can also perform subject-driven video generation (see Fig. 2), extending the scope of current image-based methods e.g., DreamboothRuiz et al. (2022) to video and motion editing. We evaluate our method extensively, demonstrating its remarkable capabilities unmatched by the baseline methods.

To summarize, our main contributions are:

1. Proposing the first method for text-based editing of real-world videos that can edit their *motion* and not merely their *appearance*.

2. Repurposing mixed training as a finetuning objective that significantly improves motion editing.

3. Presenting a new framework for text-guided image animation, by applying our video editor method on top of simple image preprocessing operations.

4. Extending the scope of subject-driven generation methods to motion generation.

## 2 RELATED WORK

### 2.1 DIFFUSION MODELS FOR SYNTHESIS

Deep diffusion models recently emerged as a powerful new paradigm for image generation Ho et al. (2020); Song et al. (2020), and have their roots in score-matching Hyvärinen & Dayan (2005); Vincent (2011); Sohl-Dickstein et al. (2015). They outperform Dhariwal & Nichol (2021) the previous state-of-the-art approach, generative adversarial networks (GANs) Goodfellow et al. (2020). While they have multiple formulations, EDM Karras et al. (2022) showed they are equivalent. Outstanding progress was made in text-to-image generation Saharia et al. (2022b); Ramesh et al. (2022); Rombach et al. (2022); Avrahami et al. (2022b), where new images are sampled conditioned on an input text prompt. Extending diffusion models to video generation is a challenging computational and algorithmic task. Early work include Ho et al. (2022c) and text-to-video extensions by Ho et al. (2022a); Singer et al. (2022). Another line of work extends synthesis to various image reconstruction tasks Saharia et al. (2022c;a); Ho et al. (2022b); Lugmayr et al. (2022); Chung et al. (2022), Horwitz & Hoshen (2022) extracts confidence intervals for reconstruction tasks.

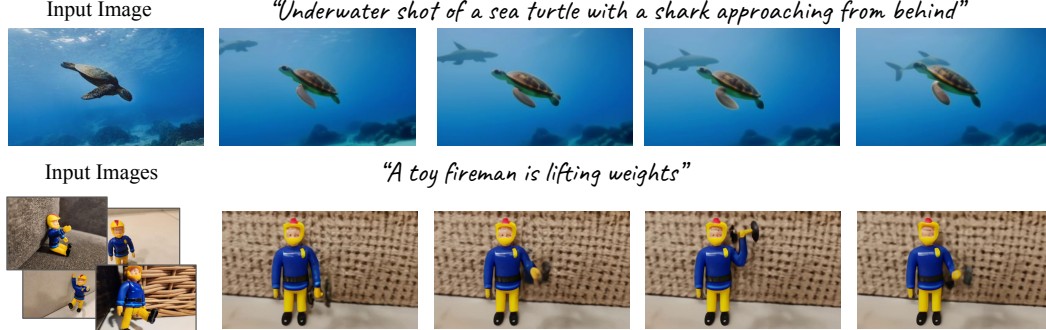

Figure 2: ***Image-to-Video editing with Dreamix:*** Dreamix instills complex motion in a static image (first row), adding a moving shark and making the turtle swim. In this case, visual fidelity to object location and background was preserved but the turtle direction was flipped. In the subject-driven case (second row), Dreamix extracts the visual features of a subject given multiple images and animates it in different scenarios such as weightlifting

## 2.2 DIFFUSION MODELS FOR EDITING

Image editing with generative models has been studied extensively, in past years many of the models were based on GANs Vinker et al. (2021); Patashnik et al. (2021); Gal et al. (2021); Roich et al. (2022); Wang et al. (2018b); Park et al. (2019); Bau et al. (2020); Skorokhodov et al. (2022); Jamriška et al. (2019); Wang et al. (2018a); Tzaban et al. (2022); Xu et al. (2022); Liu et al. (2022). Another recent line of works demonstrated preliminary generation and editing capabilities using masked image models Yu et al. (2022b); Villegas et al. (2022); Yao et al. (2021); Nash et al. (2022). However, most of the recent editing methods adopt diffusion models Avrahami et al. (2022c;a); Voynov et al. (2022). SDEdit Meng et al. (2021) proposed to add targeted noise to an input image, and then use diffusion models for reversing the process. Prompt-to-Prompt Hertz et al. (2022); Tumanyan et al. (2022); Mokady et al. (2022) perform semantic edits by mixing activations extracted with the original and target prompts. For InstructPix2Pix Brooks et al. (2022) this is only needed for constructing the training dataset. Other works (e.g. Gal et al. (2022); Ruiz et al. (2022)) use finetuning and optimization to allow for personalization of the model, learning a special token describing the content. UniTune Valevski et al. (2022) and Imagic Kawar et al. (2022) finetune on a single image, allowing better editability while maintaining good fidelity. However, the methods are image-centric and do not use temporal information. Neural Atlases Kasten et al. (2021) and Text2Live Bar-Tal et al. (2022) allow some texture-based video editing, however, unlike our method they cannot edit the *motion* of a video. A concurrent paper, Tune-a-Video Wu et al. (2022) preforms video editing by inflating a text-to-image model to learn temporal consistency. Despite their promising results, they use a text-to-image backbone that can edit video appearance *but not motion*. Their results are also not fully temporally consistent. In contrast, our method uses a text-to-video backbone, enabling *motion* editing while maintaining smoothness and temporal consistency.

## 3 BACKGROUND: VIDEO DIFFUSION MODELS

**Denoising Model Training.** Diffusion models rely on a deep denoising neural network denoted by $D_\theta$. Let us denote the ground truth video as $v$, an i.i.d Gaussian noise tensor of the same dimensions as the video as $\epsilon \sim N(0, \mathbf{I})$, and the noise level at time $s$ as $\sigma_s$. The noisy video is given by: $z_s = \gamma_s v + \sigma_s \epsilon$, where $\gamma_s = \sqrt{1 - \sigma_s^2}$. Furthermore, let us denote a conditioning text prompt as $t$ and a conditioning video $c$ (for super-resolution, $c$ is a low-resolution version of $v$). The objective of the denoising network $D_\theta$ is to recover the ground truth video $v$ given the noisy input video $z_s$, the time $s$, prompt $t$ and conditioning video $c$. The model is trained on a (large) training corpus $\mathcal{V}$ consisting of pairs of video $v$ and text prompts $t$.

**Sampling from Diffusion Models.** The key challenge in diffusion models is to use the denoiser network $D_\theta$ to sample from the distribution of videos conditioned on the text prompt $t$ and conditioning video $c$, $P(v|t, c)$. While the derivation of such sampling rule is non-trivial (see e.g. Karras

Input Video

Figure 3: *Video Motion Editing:* Dreamix can significantly change the actions and motions of subjects in a video (e.g. making a puppy leap) while maintaining temporal consistency and preserving the unedited details

et al. (2022)), the implementation of such sampling is relatively simple in practice. We follow Ho et al. (2022a) in using stochastic DDIM sampling. At a heuristic level, at each step, we first use the denoiser network to estimate the noise. We then remove a fraction of the estimated noise and finally add randomly generated Gaussian noise, with magnitude corresponding to half of the removed noise.

**Cascaded Video Diffusion Models.** Training high-resolution text-to-video models is very challenging due to the high computational complexity. Several diffusion models overcome this by using cascaded architectures. We use a model that follows the architecture of Ho et al. (2022a), which consists of a cascade of 7 models. The base model maps the input text prompt into a 5-second video of $24 \times 40 \times 16$ frames. It is then followed by 3 spatial super-resolution models and 3 temporal super-resolution models. For implementation details, see Appendix C.

## 4 EDITING BY VIDEO DIFFUSION MODELS

We propose a new method for video editing using text-guided video diffusion models. We extended it to image animation in Sec. 5.

### 4.1 VIDEO EDITING BY INVERTING CORRUPTIONS

We wish to edit an input video using the guidance of a text prompt $t$ describing the video **after** the edit. In order to do so we leverage the power of a cascade of VDMs. The key idea is to first corrupt the video by downsampling followed by adding noise. We then apply the sampling process of the cascaded diffusion models from the time step corresponding to the noise level, conditioned on $t$, which upscales the video to the final spatio-temporal resolution. The effect is that the VDM will use the low-resolution details provided by the degraded input video, but synthesize new high spatio-temporal resolution information using the text prompt guidance. While this procedure is essentially a text-guided version of SDEdit Meng et al. (2021), for complex edits e.g., *motion editing* this by itself does not result in sufficiently high-fidelity videos. To mitigate this issue, we use a mixed-finetuning objective described in Sec. 4.2.

**Input Video Degradation.** We downsample the input video to the resolution of the base model (16 frames of $24 \times 40$). We then add i.i.d Gaussian noise with variance $\sigma_s^2$ to further corrupt the input video. The noise strength is equivalent to time $s$ in the diffusion process of the base model. For $s = 0$, no noise is added, while for $s = 1$, the video is replaced by pure Gaussian noise. Note, that even when no noise is added, the input video is highly corrupted due to the extreme downsampling ratio.

**Text-Guided Corruption Inversion.** We can now use the cascaded VDMs to map the corrupted, low-resolution video into a high-resolution video that aligns with the text. The core idea here is that given a noisy, very low spatio-temporal resolution video, there are many perfectly feasible, high-resolution videos that correspond to it. We use the target text prompt $t$ to select the feasible outputs that not only correspond to the low-resolution of the original video but are also aligned to

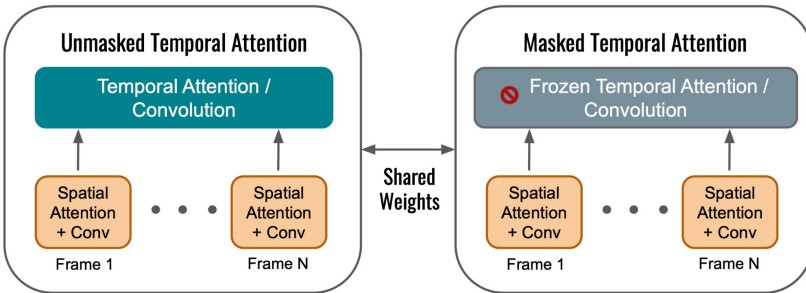

**Mixed Video-Image Finetuning**

Figure 4: ***Mixed Video-Image Finetuning:*** Finetuning the VDM on the input video alone limits the extent of motion change. Instead, we use a mixed objective that beside the original objective (bottom left) also finetunes on the unordered set of frames. We use "masked temporal attention" to prevent the temporal attention and convolution from changing (bottom right). This allows adding motion to a static video

edits desired by the user. The base model starts with the corrupted video, which has the same noise as the diffusion process at time $s$. We use the model to reverse the diffusion process up to time $0$. We then upscale the video through the entire cascade of super-resolution models (see Appendix C). All models are conditioned on the prompt $t$.

## 4.2 MIXED VIDEO-IMAGE FINETUNING

The naive method presented in Sec. 4.1 relies on a corrupted version of the input video which does not include enough information to preserve high-resolution details such as fine textures or object identity. We tackle this by adding a preliminary stage of finetuning the model on the input video $v$. Note that this only needs to be done once for the video, which can then be edited by many prompts without further finetuning. We would like the model to separately update its prior both on the appearance and the motion of the input video. Our approach therefore treats the input video, both as a single video clip and as an unordered set of $M$ frames, denoted by $u = \{x_1, x_2, .., x_M\}$. We use a rare string $t^*$ as the text prompt, following Ruiz et al. (2022). We finetune the denoising models by a combination of two objectives. The first objective updates the model prior on both motion and appearance by requiring it to reconstruct the input video $v$ given its noisy versions $z_s$.

$$\mathcal{L}_\theta^{vid}(v) = \mathbb{E}_{\epsilon \sim N(0,\mathbf{I}), s \in \mathcal{U}(0,1)} \|D_{\theta'}(z_s, s, t^*, c) - v\|^2 \tag{1}$$

Additionally, we train the model to reconstruct each of the frames individually given their noisy version. This enhances the appearance prior of the model, separately from the motion. Technically, the model is trained on a sequence of frames $u$ by replacing the temporal attention layers by trivial fixed masks ensuring the model only pays attention within each frame, and also by masking the residual temporal convolution blocks. We denote the attention masked denoising model as $D_\theta^a$. The masked attention objective is:

$$\mathcal{L}_\theta^{frame}(u) = \mathbb{E}_{\epsilon \sim N(0,\mathbf{I}), s \in \mathcal{U}(0,1)} \|D_{\theta'}^a(z_s, s, t^*, c) - u\|^2 \tag{2}$$

We train the joint objective:

$$\theta = arg \min_{\theta'} \alpha \mathcal{L}_{\theta'}^{vid}(v) + (1 - \alpha)\mathcal{L}_{\theta'}^{frame}(u) \tag{3}$$

Where $\alpha$ is a constant factor, see Fig. 4. Training on a single video or a handful of frames can easily lead to overfitting, reducing the editing ability of the original model. To mitigate overfitting, we use a small number of finetuning iterations and a low learning rate (see Appendix C). Note that while such a training objective was used by Imagen-VideoHo et al. (2022a) and VDMHo et al. (2022c), its purpose was different. There, the aim was to increase dataset size and diversity by training on large image datasets. Here, the aim is to enforce the style of the video in the model, while allowing motion editing.

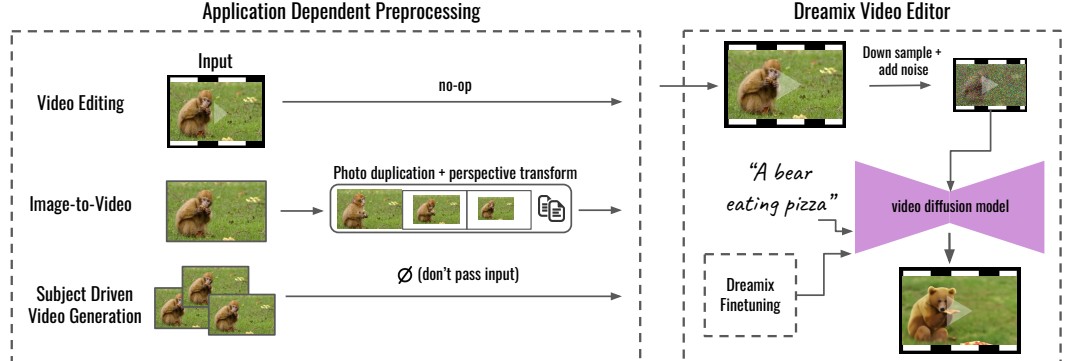

Figure 5: *Inference Overview:* Our method supports multiple applications by converting the input into a uniform video format (left). For image-to-video, the input image is duplicated and transformed using perspective transformations, synthesizing a coarse video with some camera motion. For subject-driven video generation, the input is omitted - finetuning alone takes care of the fidelity. This coarse video is then edited using our general "Dreamix Video Editor" (right): we first corrupt the video by downsampling followed by adding noise. We then apply the finetuned text-guided VDM, which upscales the video to the final spatio-temporal resolution

## 5 APPLICATIONS OF DREAMIX

The method proposed in Sec. 4, can edit motion and appearance in real-world videos. In this section, we propose a framework for using our Dreamix video editor for general, text-conditioned *image-to-video* editing, see Fig. 5 for an overview.

**Dreamix for Single Images.** Provided our general video editing method, Dreamix, we now propose a framework for image animation conditioned on a text prompt. The idea is to transform the image or a set of images into a coarse, corrupted video and edit it using Dreamix. For example, given a single image $x$ as input, we can transform it to a video by replicating it 16 times to form a static video $v = [x, x, x...x]$. We can then edit its appearance and motion using Dreamix conditioned on a text prompt. Here, we do not wish to incorporate the motion of the input video (as it is static and meaningless) and therefore use only the masked temporal attention finetuning ($\alpha = 0$). To create "cinematic" effects, we can further control the output video by simulating camera motion, such as panning and zoom. We perform this by sampling a smooth sequence of 16 perspective transformations $T_1, T_2..T_{16}$ and apply each on the original image. When the perspective requires pixels outside the input image, we simply outpaint them using reflection padding. We concatenate the sequence of transformed images into a low quality input video $v = [T_1(x), T_2(x)..T_{16}(x)]$. While this does not result in realistic video, Dreamix can transform it into a high-quality edited video. See Appendix D for details on the applied transformations.

**Dreamix for subject-driven video generation.** We propose to use Dreamix for text-conditioned video generation given an image collection. Differently from existing methods, e.g., Dreambooth Ruiz et al. (2022), it can add motion and not only change appearance. The input to our method is a set of images, each containing the subject of interest. This can also use different frames from the same video, as long as they show the same subject. Higher diversity of viewing angles and backgrounds is beneficial for the performance of the method. We then use the finetuning method from Sec. 4.2, where we only use the masked attention finetuning ($\alpha = 0$). After finetuning, we use the text-to-video model *without* a conditioning video, but rather only using a text prompt (which includes the special token $t^*$).

## 6 EXPERIMENTS

In this section, we establish that Dreamix is able to edit motion in real-world videos and images, a major improvement over the existing methods. To fully experience our results, please see the supplementary videos.

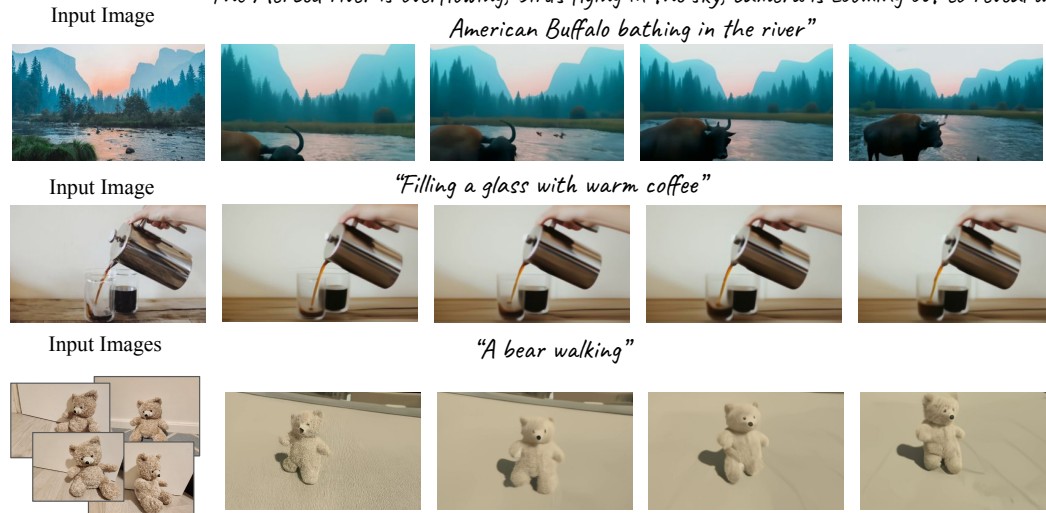

Figure 6: ***Additional Image-to-Video Results:*** First row - the image is zoomed out to reveal a bathing buffalo. Dreamix can also instill motion in a static image as in the second row where the glass is gradually filled with coffee. Third row - animating a subject based on a small number of independent images

Table 1: ***User Study:*** Users rated editing results by quality, fidelity to the base video and alignment with the text prompt. Based on visual inspection, we require an edit to score greater than 2.5 in all dimensions to be successful and observe that Dreamix is the only method to achieve the desired trade off

| Method | Quality | Fidelity | Alignment | Success |
|--------|---------|----------|-----------|---------|
| PnP | 2.16 $\pm$1.13 | 3.78 $\pm$0.99 | 3.39 $\pm$1.38 | 20% |
| TaVid | 1.99 $\pm$0.92 | 3.29 $\pm$1.21 | 2.69 $\pm$1.55 | 13% |
| Ours | 3.58 $\pm$1.04 | 3.55 $\pm$1.09 | 3.79 $\pm$1.33 | 76% |
| Uncond. | 3.43 $\pm$1.09 | 2.49 $\pm$1.12 | 4.28 $\pm$1.02 | 45% |

## 6.1 QUALITATIVE RESULTS

**Video Editing.** In Fig. 1, we change the motion to dancing and the appearance from monkey to bear while keeping the coarse attributes of the video fixed. Dreamix can also generate new motion that does not necessarily align with the input video (puppy in Fig. 3, orangutan in supplementary material (SM)), and can control camera movements (zoom-out example in the SM). Dreamix can generate smooth visual modifications that align with the temporal information in the input video. This includes adding effects (field and saxophone in the SM), adding or replacing objects (hat, skateboard, and robot in the SM), and changing the background (truck in the SM).

**Image-driven Videos.** When the input is a single image, Dreamix can use its video prior to add new moving objects (camel in SM), inject motion into the input (turtle in Fig. 2 and coffee in Fig. 6), or new camera movements (buffalo in Fig. 6). Although Singer et al. (2022); Yu et al. (2022b) perform image-driven animations, they can only add very simple motions (e.g. animating water or snowfall). Our method is unique in adding large motions and moving objects into general, real-world images.

**Subject-driven Video Generation.** Dreamix can take an image collection showing the same subject and generate new videos with this subject in motion. This is unique, as previous approaches could only output still images. We demonstrate this on a range of subjects and actions including: the weight-lifting toy fireman in Fig. 2, walking and drinking bear in Fig. 6 and SM. It can place the subjects in new surroundings, e.g., moving caterpillar on a leaf or even under a magnifying glass (see SM).

Table 2: ***Baseline Comparisons:*** Our method achieves better temporal consistency than PnP and Tune-a-Video (TaVid). Moreover, Dreamix is successful at *motion* editing while other methods cannot. This is reflected in the better quality (low LPIPS) and alignment (high CLIP Score). While the unconditional method seems to outperform Dreamix, it has poor fidelity as it is not conditioned on the input video

| Metric | PnP | TaVid | Ours | Uncond. |
|---|---|---|---|---|
| LPIPS ↓ | 0.209 | 0.145 | 0.112 | 0.101 |
| CLIP Score ↑ | 0.304 | 0.303 | 0.317 | 0.320 |
| Fidelity | See user study (Tab. 1) for evaluation | | | |

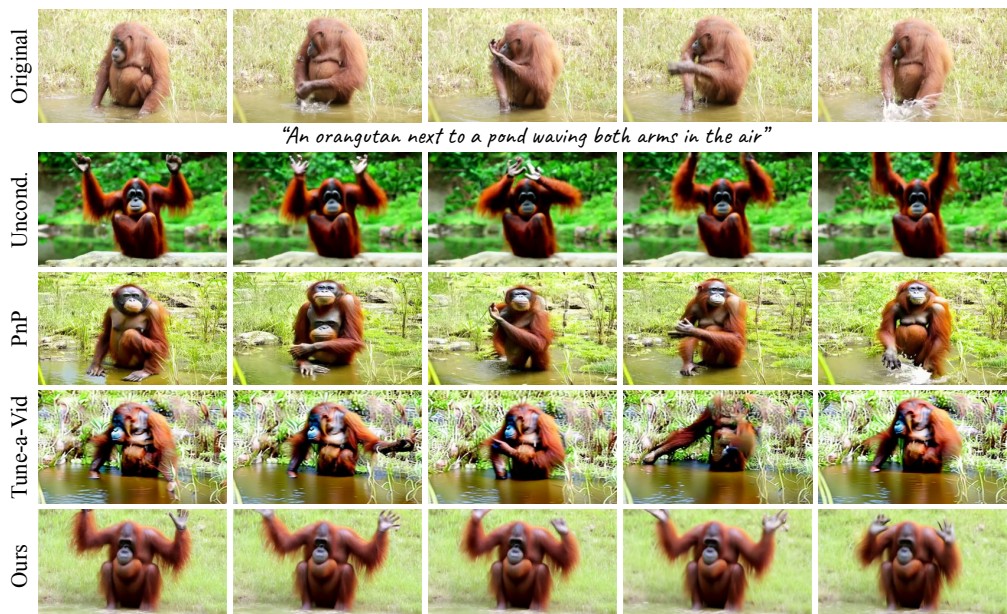

Figure 7: ***Comparison to Baseline Methods for Motion Edits:*** Although the quality and alignment of unconditional generation are high, there is no resemblance to the original video (low fidelity). While PnP and Tune-a-Video preserve the scene, they fail to edit the motion according to the prompt (no waving) and suffer from poor temporal consistency (flickering). Our method is able to edit the *motion* according to the prompt while preserving the fidelity and generating a high quality video. Moreover, video-based methods (Uncond. and ours) exhibit motion blur, also present in real videos

## 6.2 BASELINE COMPARISONS

**Baselines.** We compare our method against three baselines: *Unconditional.* Directly mapping the text prompt to a video, without conditioning on the input video using a model similar to Imagen-Video. *Plug-and-Play (PnP).* Applying PnP Tumanyan et al. (2022) on each video frame independently. *Tune-a-Video (TaVid).* Finetuning Tune-a-Video Wu et al. (2022) on the input video.

**Data.** We created a dataset of 29 videos taken from YouTube-8M Abu-El-Haija et al. (2016) and 127 text prompts, spanning different categories (see Appendix E).

**Quantitative Comparison.** We measure alignment by the frame-level CLIP Score Hessel et al. (2021) and quality (stability) with LPIPS Zhang et al. (2018) between consecutive frames. As automatic metrics do not measure fidelity and are imperfectly aligned with human judgement, we also conduct a user study. A panel of 20 evaluators rated each video/prompt pair on a scale of $1 - 5$ to evaluate its quality, fidelity and alignment. When visually inspecting the results we discover that videos that received a score lower than $2.5$ in any of the dimensions are usually clear failure cases. Therefore we also report the percentage of items where all dimensions are larger than $2.5$ (i.e. "Success"). See Appendix F.2 for additional details on the evaluation protocol.

Table 3: ***Ablation Study:*** Left: Users were asked to compare text-guided video edits of with (w/ Ft) and without (w/o Ft) finetuning. "None" indicates failure of both methods according to user. Apart from style-based edits, where high fidelity is not needed, finetuning significantly improves the results. Right: Users were asked to compare video finetuning (Vid) with mixed video-image finetuning (Mix). Mixed finetuning significantly improves the results for most cases

| Edit Type | # of Edits | w/o Ft. | w/ Ft. | None | Vid | Mix |
|---|---|---|---|---|---|---|
| Motion | 36 | 17% | 72% | 11% | 35% | 65% |
| Object | 44 | 36% | 48% | 16% | 62% | 38% |
| Background | 32 | 19% | 77% | 9% | 36% | 64% |
| Style | 15 | 67% | 27% | 6% | 26% | 74% |

The evaluation and user study are presented in Tab. 2 and Tab. 1. Image-based methods (PnP, Tune-a-Video) exhibit impaired temporal consistency, resulting in low quality. Moreover, they are unable to perform *motion* edits, resulting in poor alignment and high fidelity. Video-based methods maintain temporal consistency while allowing motion editing. Although unconditional generation outperforms our method in the automatic evaluations (Tab. 2), it has poor fidelity (Tab. 1) as it is not conditioned on the input video. Overall, our method has the highest success rate.

**Qualitative Comparison.** Figure 7 presents an example of motion editing by Dreamix compared to the baselines. The text-to-video model achieves low fidelity edits as it is not conditioned on the original video. PnP preserves the scene but fails to perform the edit and lacks consistency between different frames. Tune-a-Video exhibits better temporal consistency but still fails to perform the motion edit. Dreamix performs well on all three objectives, adding the desired motion while preserving fidelity and high-quality.

## 6.3 ABLATION STUDY

We ablate the use of finetuning and the mixed video-image finetuning by performing a user study using the dataset described above. The ablation indeed supports the idea of using finetuning in cases where high-editability is required. We can see that *Motion* changes require high-editability and are thus improved by finetuning. Moreover, as the noising corrupts the video, preserving fine-details in *background, color or texture* edits requires finetuning. In contrast, denoising without finetuning worked well for *style* edits, where finetuning was often detrimental. This is expected as *style* edits are often conflicted with high fidelity preservation (e.g. changing the texture of an object means reducing fidelity). The ablation shows that in most cases mixed finetuning improves the results by a wide margin. Results are presented in Tab. 3, a visual ablation is found in Appendix F.2.

## 7 LIMITATIONS

While Dreamix is the first diffusion-based video method that can edit motion, it has limitations.

**Computational Cost.** VDMs are computationally expensive. Finetuning our model using 4 TPU v4 accelerators requires around 30 minutes per video. Once finetuned, sampling takes roughly 2 minutes on similar hardware. Speeding it up will allow Dreamix to be used for more applications.

**Comparison to Image-based Methods.** Dreamix uses VDMs while previous approaches used image-level methods. As VDMs are nascent and have lower resolution than image DMs, this presents an interesting trade-off. Dreamix has the ability to edit motion and has high temporal consistency, while previous methods e.g., PnP and Tune-a-Video, can have higher spatial resolution. Although Tune-a-Video can achieve high alignment for texture editing on videos with limited motion, it suffers from poor temporal consistency (see SM). This highlights the importance of using a VDM backbone that provides temporal consistency and enables *motion* editing.

## 8 CONCLUSION

We presented the first diffusion-based method that can edit motion in real-world videos. Our method can be applied to image animation and subject-driven video generation. Extensive experiments demonstrated the unprecedented capabilities of our method.

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

## A    ATTACHED VIDEOS

In addition to this appendix, we include a number of videos, **we highly encourage the reviewer to view them**. The included videos are:

1. "1_dreamix_overview_video.mp4" - An overview video of our method with audio narration.
2. "2_dreamix_video_editing_examples.mp4" - A number of video editing examples generated by our method (Dreamix).
3. "3_dreamix_image2video_examples.mp4" - A number of image-to-video examples generated by our method (Dreamix).
4. "4_dreamix_subject_driven_video_generation_examples.mp4" - A number of subject-driven video generation examples generated by our method (Dreamix).
5. "5_dreamix_baseline_comparisons.mp4" - A number of videos comparing our method (Dreamix) to the other baselines.

**Note: to match the conference requirement of maximum $100$MB for the supplementary, all the videos are compressed, the uncompressed versions will be released in the final revision**.

## B    SOCIAL IMPACT

Our primary aim in this work is to advance research on tools to enable users to animate their personal content. While the development of end-user applications is out of the scope of this work, we recognize both the opportunities and risks that may follow from our contributions. As discussed above, we anticipate multiple possible applications for this work that have the potential to augment and extend creative practices. The personalized component of our approach brings particular promise as it will enable users to better align content with their intent, despite potential biases present in general VDMs. On the other hand, our method carries similar risks as other highly capable media generation approaches. Malicious parties may try to use edited videos to mis-lead viewers or to engage in targeted harassment. Future research must continue investigating these concerns.

## C    IMPLEMENTATION DETAILS

### C.1    ARCHITECTURE

All of our experiments were preformed on a VDM that is similar to Imagen-Video Ho et al. (2022a), a pertrained cascaded text-to-video diffusion model, with the following components:

1. A T5-XXLRaffel et al. (2020) text encoder, that computes embeddings from the textual prompt. This embeddings are then used as conditioning by all other models.
2. A base video diffusion model, conditioned on text. It generates videos at $16 \times 24 \times 40 \times 3$ resolution (frames $X$ height $X$ width $X$ channels) at 3 fps.
3. 6 super-resolution video diffusion models, each conditioned on the text and the output video of the previous model. Each model is either spatial (SSR), i.e. upscales resolution, or temporal (TSR), i.e. fills in intermediate frames between the input frames. The order of super resolution models is TSR (2x), SSR (2x), SSR(4x), TSR(2x), TSR(2x), and SSR(4x). The multiplier in the parenthesis for output frames (for TSR), and for output pixels in height and width (for SSR). The final output video is in $128 \times 768 \times 1280 \times 3$ at 24 fps.

Note that the diffusion models are pretrained on both videos and images, with frozen temporal attention and convolution for the latter. Our mixed finetuning approach treats video frames as if they were images.

**Distillation.** For some of these models, we use a distilled version to allow for faster sampling times. The base model is a distilled model with $64$ sampling steps. The first two SSR models are non-distilled models with $128$ sampling steps (due to finetuning considerations, see below). All other SR models use $8$ sampling steps. All models use classifier-free-guidance weight of 1.0 (meaning that classifier free guidance is turned off).

## C.2  Finetuning

To reduce finetuning time, we only finetune the base model and the first 2 SSR models. In our experiments, finetuning the first 2 SSR models using the distilled models (with 8 sampling steps) did not yield good quality. We therefore use the non-distilled versions of these models for all experiments (including non-finetuned experiments). When using "Mixed Video-Image Finetuning" we use $\alpha = 0.35$, and finetune for 300 steps. For all our experiments we use a learning rate of $6 \cdot 10^{-6}$.

## C.3  Sampling

We use a DDIM sampler with stochastic noise correction, following Ho et al. (2022a). For the last highest resolution SSR, for capacity reasons, we use the model to sample a sub-chunks of 32 frames of the input lower resolution videos, and then we concatenate all the outputs together back to 128 frame videos.

## D  Image-to-Video Transformations

We only use perspective transformations to create "cinematic" effects, e.g., panning, zooming, and camera shake. In our supplementary, we included Image-to-Video examples with different perspective transformations applied to them. We detail these transformations in Tab. 4. Some of the examples did not use the perspective transformations at all. Also, ensuring the smoothness of the transformed sequence is unnecessary as this is fixed by the diffusion and super-resolution processes.

Table 4: *Perspective Transformations*

| Video | Timestamp | Transformation | Effect |
|---|---|---|---|
| Plant | 00:00 | Translate | Pan |
| Turtle | 00:11 | Rand. translate | Shake |
| Coffee | 00:22 | Translate | Pan |
| Camel | 00:33 | None | None |
| Volcano | 00:43 | Rand. translate | Shake |
| Bear | 00:54 | Perspective | Pan |
| Penguins | 01:05 | None | None |
| Unicorn | 01:15 | Scale | Zoom out |
| Buffalo | 01:26 | Scale | Zoom out |
| Bigfoot | 01:37 | Translate | Pan |

## E  Evaluation dataset

In all evaluations described in the paper, we used a dataset of 29 videos with 127 edit prompts. The dataset videos were selected from YouTube-8M Abu-El-Haija et al. (2016) and show animals, people performing actions, vehicles, and other objects. The edit prompt categories are motion, object, background, and style. In the motion category the prompts perform motion editing (e.g. adding motion with the prompt "An orangutan next to a pond waving both arms in the air"), the object category performs object level edits (e.g. adding a party hat with the prompt "A puppy walking with a party hat"), the background category performs edits of the background (e.g. adding a river with the prompt "A blue pickup truck crossing a deep river"), and the style category performs style-transfer like edits (e.g. changing the style to cartoon with the prompt "A cartoon of a man playing a saxophone").

## F  Human evaluation details

We performed human evaluations for the baseline comparison and the ablation analysis. The evaluations were conducted by a panel of 20 human raters using the dataset described in **??**. The video

resolution shown to raters was $350 \times 200$, except for tune-a-video where we used a resolution of $200 \times 200$ (because we observed it performs better with square outputs).

## F.1 ABLATION STUDY

In the ablation study the raters were asked to select the best edited video out of 12 hyperparameter combinations:

- no finetuning, with $s \in 0.4, 0.7, 0.8, 0.85$

- video finetuning for 64 steps, with $s \in 0.8, 0.9, 0.95, 0.98$

- mixed finetuning, with $(ft_{steps}, s) \in (150, 0.98), (200, 0.98), (200, 1.0), (300, 1.0)$

A visual example of the tradeoff between the amount of noise and the amount of finetuning is shown in Fig. 8.

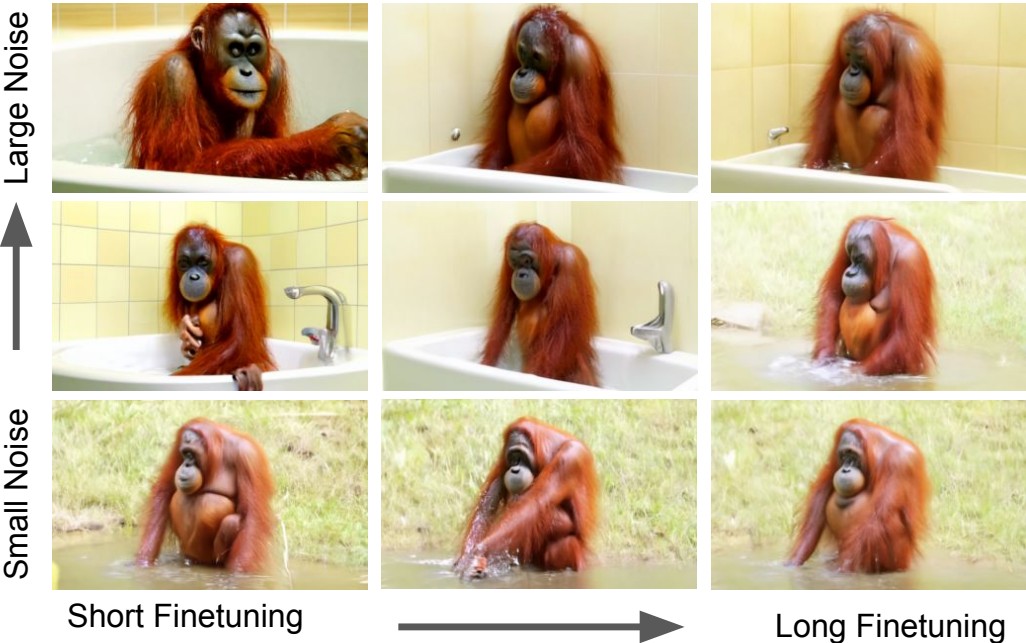

Figure 8: *Noise-Finetuning Tradeoffs:* We compare the effect of noise magnitude and number of finetuning iterations on edited videos. The original frame is on the bottom left, the rest were generated by different parameters for the prompt "An orangutan with orange hair bathing in a bathroom". We can observe that higher noise allows for larger edits but reduces fidelity. More finetuning iterations improve fidelity at higher noises. The best results are obtained for high noise and a large number of finetuning iterations

## F.2 BASELINE COMPARISON

In the baseline comparisons described in the paper, raters evaluated videos on quality, fidelity and alignment. The raters saw the original video alongside an edited video and answered the following questions on a scale of $1 - 5$:

1. "Rate the overall visual quality and smoothness of the edited video."

2. "How well does the edited video match the textual edit description provided?"

3. "How well does the edited video preserve unedited details of the original video?"

## F.3 DIRECT COMPARISON

We also conducted a direct comparisons between the different editing methods. In this comparison raters saw the videos simultaneously and selected the best edit. We conducted the comparison once with a fixed set of hyperparameters for Dreamix, and once more showing a single Dreamix video chosen among 12 hyperparameters sets. Results can be seen in Tab. 5.

Table 5: **_Direct comparison of editing methods:_** Users were shown editing results of different editing methods were asked to pick the best one. In the "Multiple HP" column, the Dreamix video was chosen from a set of 12 hyperparameters. We show the number of times each method got a majority vote (out of 5 ratings)

| Method | Single HP | Multiple HP |
|---|---|---|
| Plug-and-Play | 2% | 1% |
| Tune-a-Video | 6% | 6% |
| Ours | 34% | 77% |
| No good edit / Uncond. | 58% | 16% |

## G QUANTITATIVE EVALUATION

In the quantitative baseline comparison described in the paper we reported alignment and quality. We measure alignment by the frame-level CLIP Score Hessel et al. (2021). That is, we compute the cosine similarity between the CLIP Radford et al. (2021) embedding and the CLIP text embedding for each frame. For each video we take the average over all frames, finally we report the mean over all the videos. For quality (stability) we compute the LPIPS Zhang et al. (2018) distance between all pairs of consecutive frames. For each video we take the average over all pairs of consecutive frames, finally we report the mean over all the videos. To perform a fair comparison with Tune-a-Video (which outputs videos of 24 frames at 5 fps) we subsampled the rest of the methods to match this framerate. Additionally, before passing through CLIP and LPIPS, all the frames are preprocessed to match the required format (i.e. resize to 224, center crop to 224, ImageNet normalization).

## H IMAGE ATTRIBUTION

- Desert - https://unsplash.com/photos/PP8Escz15d8
- Fuji mountain https://unsplash.com/photos/9Qwbfa_RM94
- Tree in snow - https://unsplash.com/photos/aQNy0za7x0k
- Hut in snow - https://unsplash.com/photos/qV2p17GHKbs
- Lake with trees - https://unsplash.com/photos/dIQlgwq6V3Y
- Plant - https://unsplash.com/photos/LrPKL7jOldI
- Turtle - https://unsplash.com/photos/za9MCg787eI
- Yosemite - https://unsplash.com/photos/NRQV-hBF10M
- Foggy forest - https://unsplash.com/photos/pKNqyx_v62s
- Coffee - https://unsplash.com/photos/SMPe5xfbPT0
- Monkey - https://www.pexels.com/video/a-brown-monkey-eating-bread-2436088/

## I ADDITIONAL RESULTS

Below we present additional results of our method, for the best experience see the included videos.

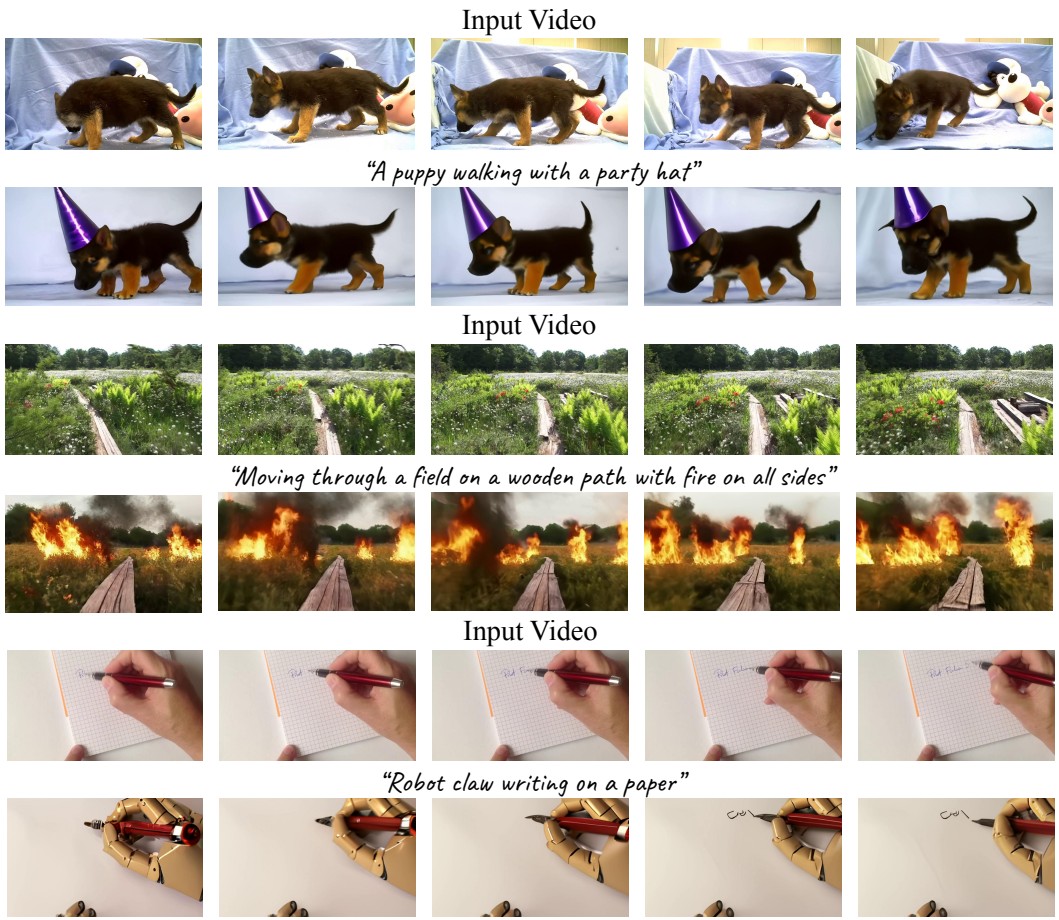

Figure 9: *Additional Video Editing Results (1/5)*

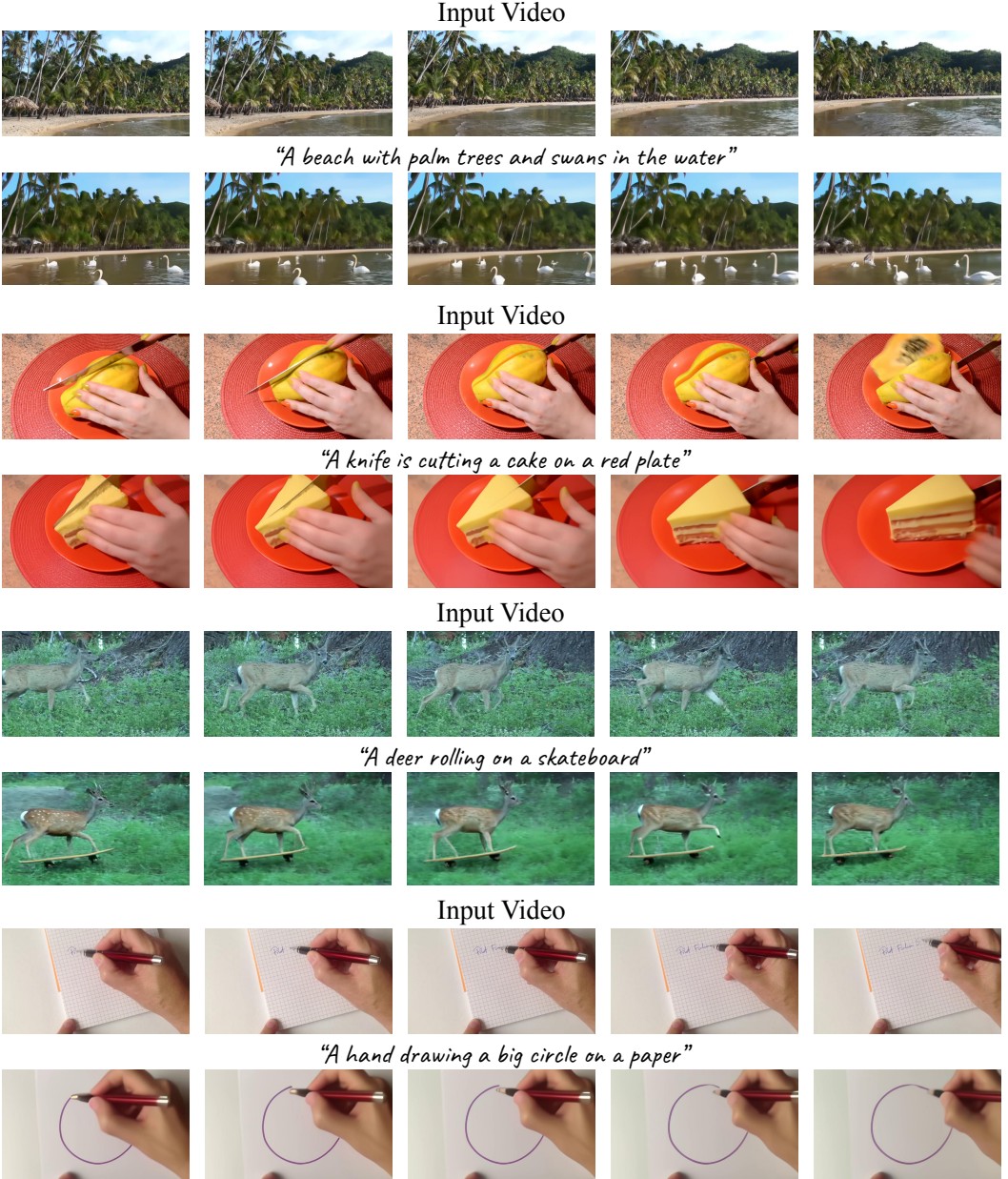

Figure 10: *Additional Video Editing Examples (2/5)*

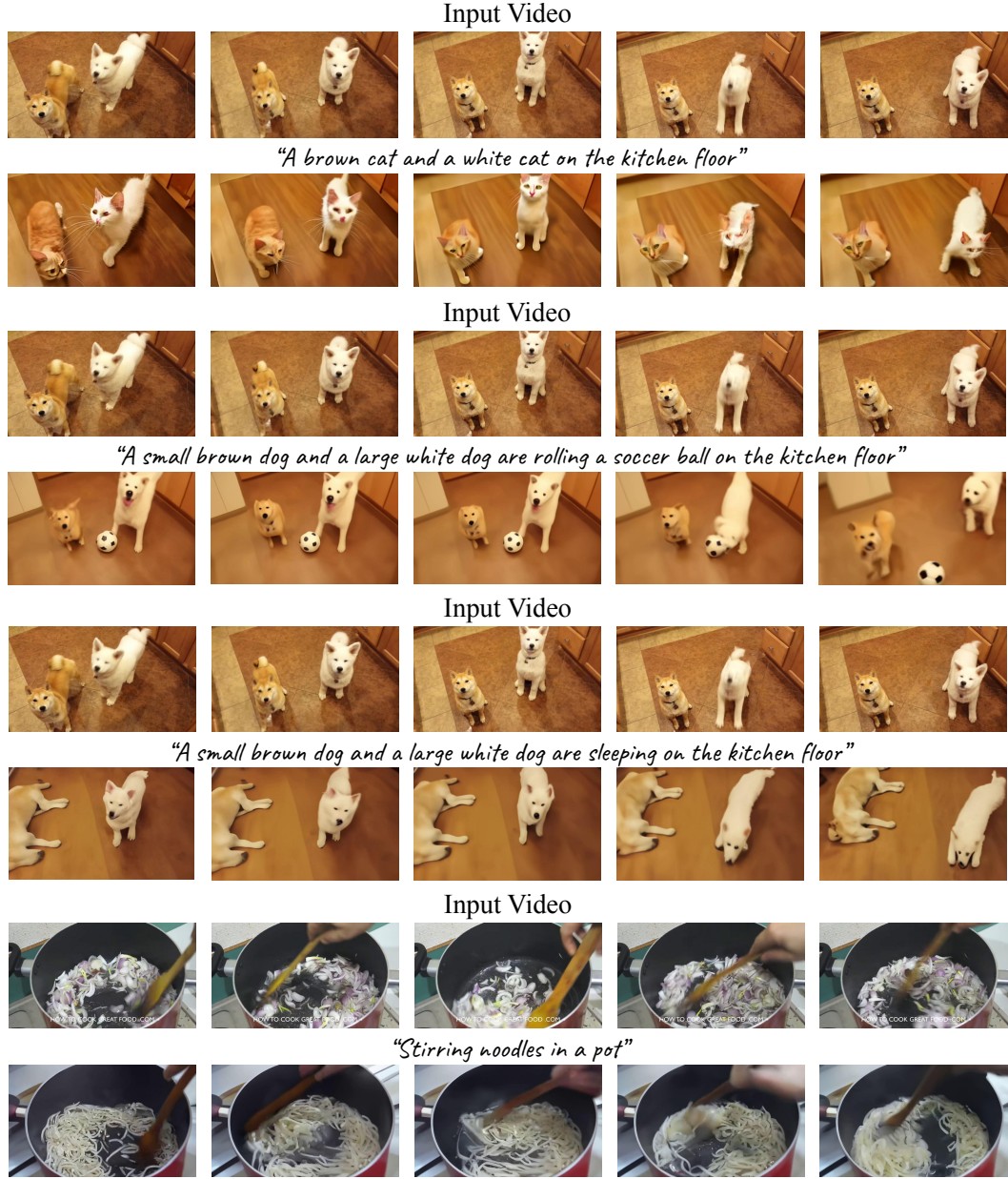

Figure 11: *Additional Video Editing Examples (3/5)*

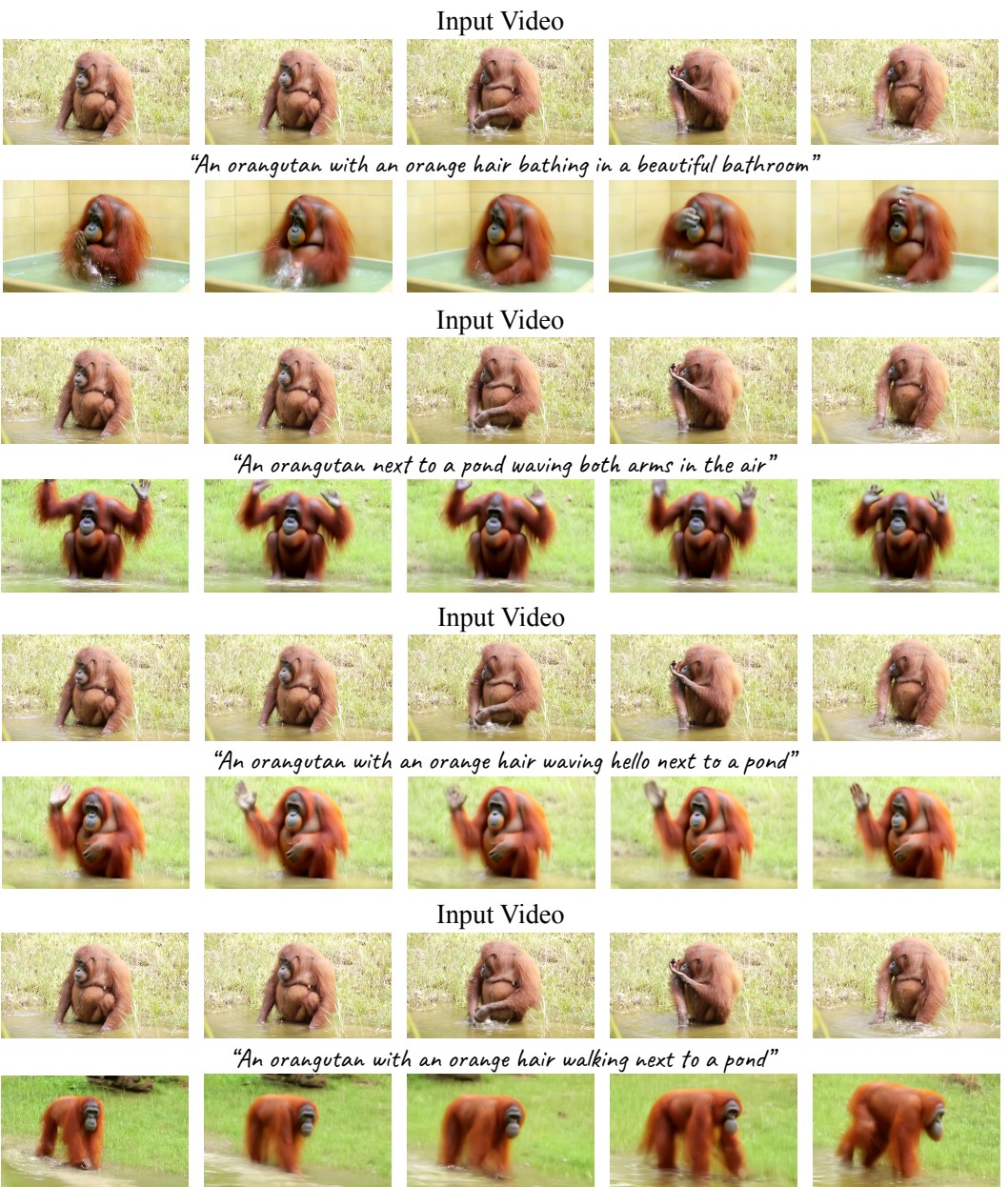

Figure 12: *Additional Video Editing Examples (4/5)*

Input Video

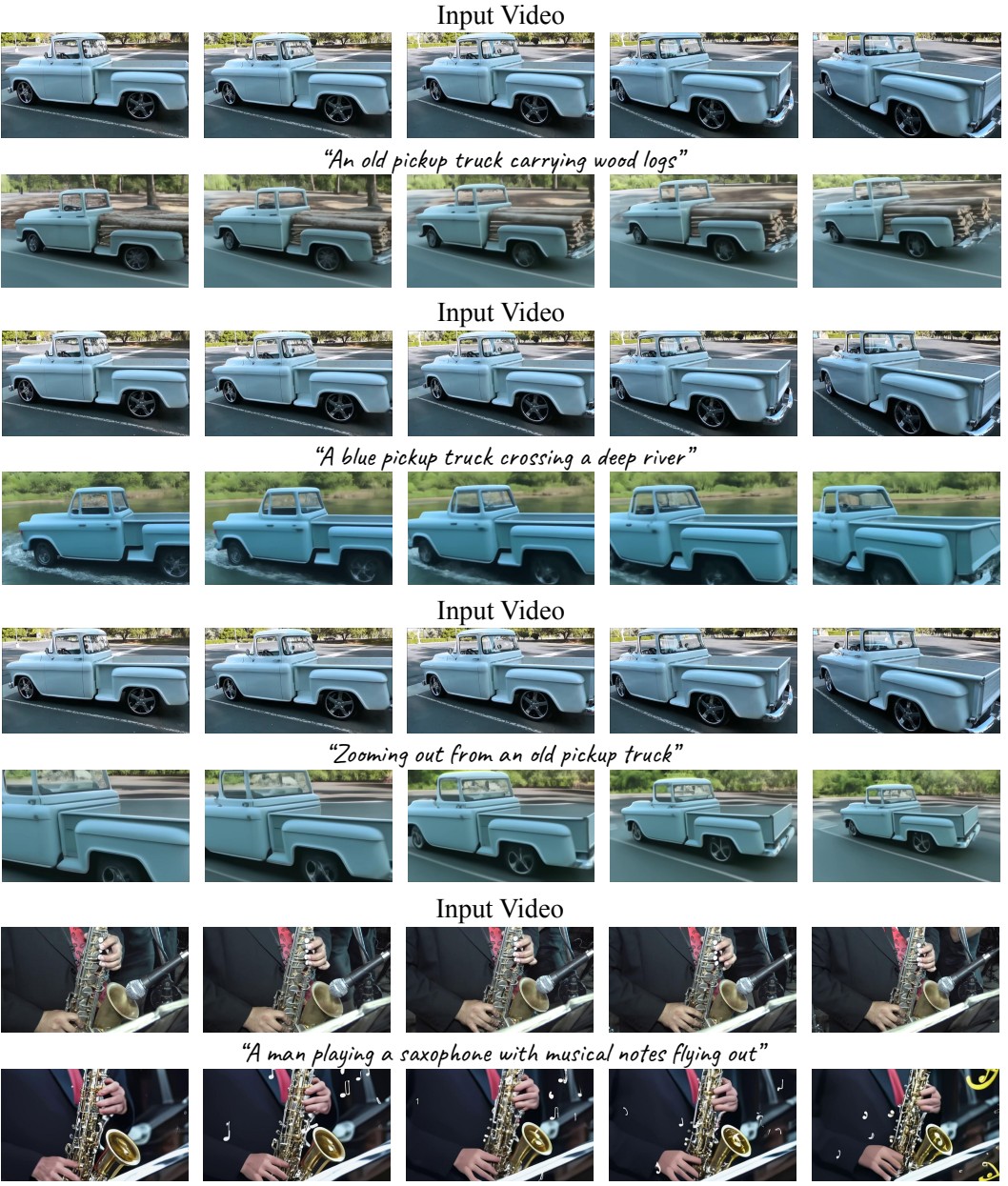

"An old pickup truck carrying wood logs"

Input Video

"A blue pickup truck crossing a deep river"

Input Video

"Zooming out from an old pickup truck"

Input Video

"A man playing a saxophone with musical notes flying out"

Figure 13: *Additional Video Editing Examples (5/5)*

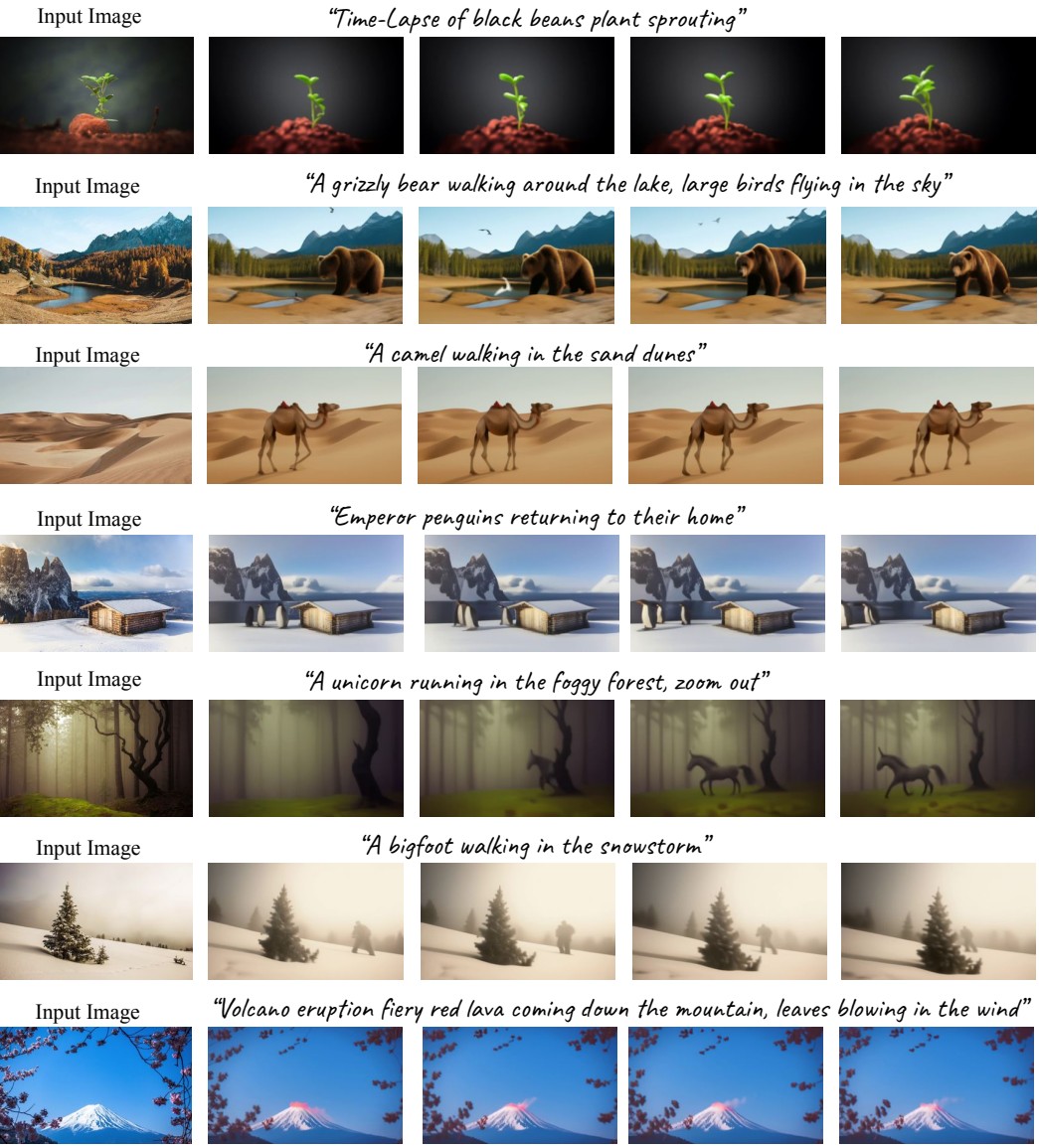

Figure 14: *Additional Image-to-Video Examples*

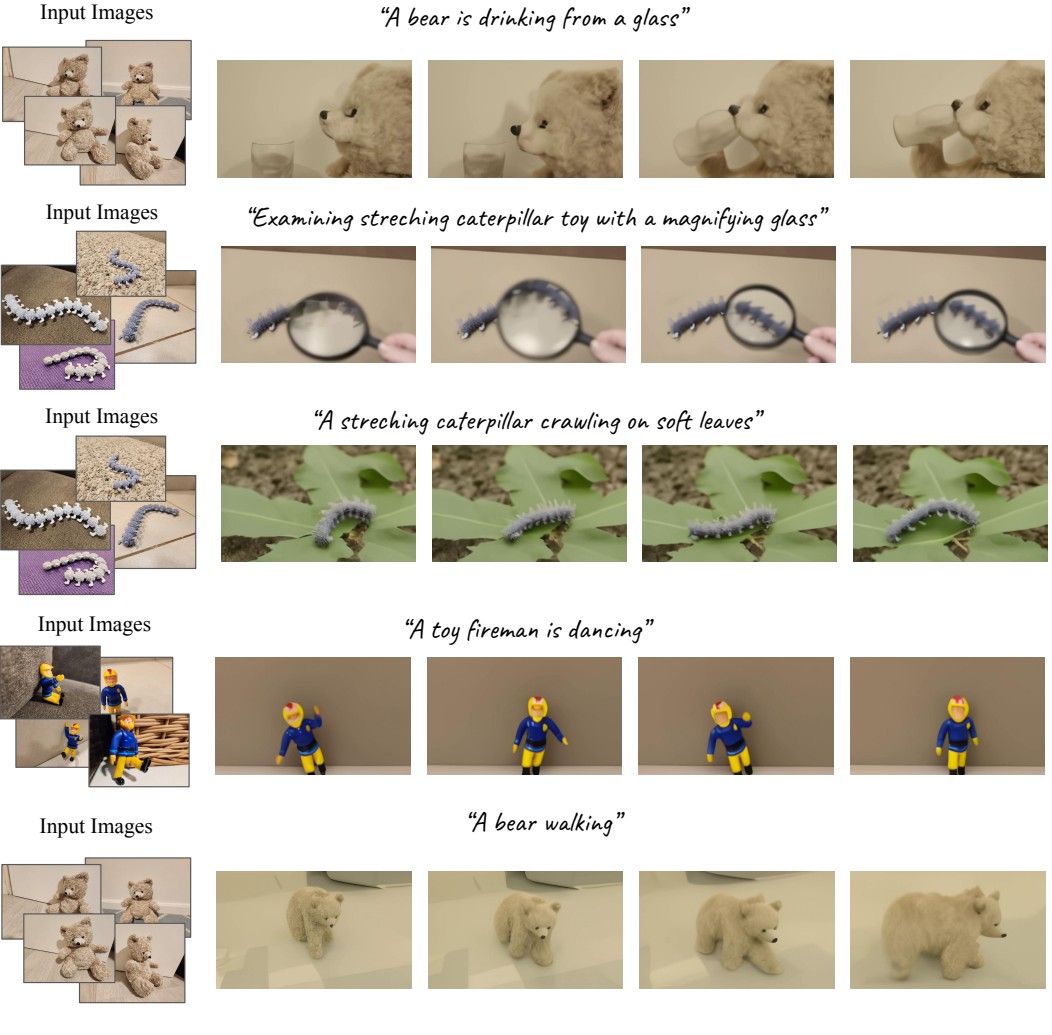

Figure 15: *Additional Subject-Driven Video Generation*

