# OpenReview forum: "Dreamix: Video Diffusion Models are General Video Editors"
_ICLR.cc/2024/Conference — ICLR 2024 Conference Withdrawn Submission_

### Official Review · Reviewer_JbPW · 2023-10-30

**Soundness:** 3 good
**Presentation:** 3 good
**Contribution:** 3 good
**Rating:** 8
**Confidence:** 5

**Summary:**

This paper proposes a text-driven motion edition method based on video diffusion models. The authors leverage a cascade of VDMs and use a mixed-finetuning objective (full temporal attention and masked temporal attention) for high-fidelity retention. The authors also extend their method to animation generation. Extensive experimental results show impressive creativity of the proposed method, especially in the supplementary materials.

**Strengths:**

Motion edition is a challenging and important topic. The authors propose a systematic solution for text-driven motion edition based on video diffusion models, which has a positive impact on the entire community. The authors present a good number of experiments validating the effectiveness of their approach and demonstrate excellent performance.

**Weaknesses:**

1. Limited fidelity to original videos. Ensuring the fidelity of the original video is a thorny issue. Also, in the presentation:

**Spatially** The background has become smooth and appears to be peeling (ref. Figure 7). The authors need to explain what efforts the authors have made to address this issue.

**Temporally** Lack of integration of action semantics and temporal modeling, which leads to the incongruity of target motions. *e.g.,* the transition from eating to dancing in Figure 1.

2. The proposed method is computationally expensive due to the introduction of VDMs.

**Questions:**

1. In page 5, the authors say:*Technically, the model is trained on a sequence of frames u by replacing the temporal attention layers by trivial fixed masks ensuring the model only pays attention within each frame, and also by masking the residual temporal convolution blocks.* What do you mean about temporal attention layers? Why use an unordered set of M frames? In my opinion, the former is a technical operation at the model level, while the latter is a processing at the data level. Is there any necessary connection between the two technologies?

2. Can the authors explain why extending this method to animation generation works? After all, an image cannot provide temporal information about actions, and how to ensure the continuity of video frames is a key issue.

---

> ### Author Response · Authors · 2023-11-17
>
> We sincerely appreciate the constructive review and offer the following responses:
>
> 1. **Fidelity:** We acknowledge the concern raised about the limited fidelity to original videos, particularly in terms of appearance smoothness observed in Figure 7. While our mixed finetuning strategy significantly improves fidelity, we recognize that appearance smoothness remains an area for improvement. Our focus, however, is primarily on motion generation, and we anticipate that advancements in VDM resolution will naturally enhance the appearance fidelity of Dreamix without requiring additional modifications.
>
> 2. **Computational Cost:** The reviewer correctly notes the computational expense introduced by our method due to the utilization of VDMs. We share this concern and believe that new advancements, e.g. latent consistency model ideas, will likely mitigate this computational expense of VDMs, making the method more efficient.
>
> 3. **Writing:** The reviewer is correct, we will try to clarify the writing.
>
> 4. **Animation Capability:** We hypothesize that the pretraining of the VDM endows it with robust capabilities to predict motion patterns even from a single frame, contributing to the success of our method in animation generation.
>
> We appreciate the insightful questions and concerns raised by the reviewer.

---

### Official Review · Reviewer_cmbb · 2023-10-30

**Soundness:** 3 good
**Presentation:** 3 good
**Contribution:** 3 good
**Rating:** 5
**Confidence:** 3

**Summary:**

introduce Dreamix, a video editing framework that combines low-resolution spatio-temporal information from the original video with high-resolution synthesized information aligned with a text prompt. The framework maintains fidelity to the original video by using a preliminary stage of finetuning the model on the original video. The authors propose a mixed finetuning approach that improves motion editability by training the model on both the original video and its individual frames. Additionally, Dreamix can be used to animate still images and perform subject-driven video generation. The authors provide extensive experiments to showcase the capabilities of Dreamix.

**Strengths:**

1. the proposed framework can be applied top multi-tasks like Video Editing, Image-driven Videos and Subject-driven Video Generation.

2. the Mixed Video-Image Finetuning is sound and with reasonable performance

3. The paper presents extensive experiments that demonstrate the ability of Dreamix.

**Weaknesses:**

1.  although the Mixed Video-Image Finetuning strategy is sound, the overall technical novelty is kind limited.  it is an application of VDMs with sophisticated strategy during finetuing. Also, it would be interesting to see how much would the base VDM would effect the finetuing results.

2. although the paper claims high fidelity and quality for video editing, the resolutions are still low and with blurred details.  It looks like the input video/image provides layout information and the details are generated by VDMs.

3. for Image-driven Videos, the method can not maintain the consistent between the input and first frame, and the quality is worse than more recent work like animatefiff.

4. for Subject-driven Video Generation, the appearance feature and details can not be maintained, like the   toy in figure 2 and bear in figure 6.

**Questions:**

Please check weakness

---

> ### Author Response · Authors · 2023-11-17
>
> We appreciate the thorough review provided by the reviewer and address their concerns as follows:
>
> 1. **Technical Novelty:** While we acknowledge that our mixed finetuning strategy may not be deemed highly technically novel, our primary claim is the generality of Video Diffusion Models (VDMs) as video editors. We showcase different video generation tasks and propose enhancements to an existing VDM, demonstrating, for the first time, motion edits of a video. The significance of this research therefore extends beyond the technical novelty, particularly by its insights and applicability.
>
> 2. **Fidelity:** The input, coupled with our mixed finetuning strategy, contributes to the details, as evidenced in Table 3. We openly discuss the limitation of lower resolution generation in our paper under the limitations section.
>
> 3. **Image-driven video generation:** While there is some resolution degradation between the input image and the generated results, our method is able to add motion and animate the scene in a manner that no other method can currently perform. Moreover, AnimateDiff is irrelevant for this application as it expects a “Dreambooth” like model as input, which requires a set of images instead of a single image. However, following the suggestion of reviewer “UUNx”, we conducted a detailed comparison to current methods.  VideoComposer, Gen-2, and PikaLabs. All the methods failed to add large motion or add objects to the image and animate them.
>
> 4. **Subject-driven video generation:** We acknowledge that some fine details are not fully preserved, as discussed in the paper. Attempting a comparison with AnimateDiff for subject-driven video generation did not yield satisfactory results, as AnimateDiff is tailored for small motion generation and struggles with larger motions found in our examples (e.g., walking and drinking beer).
>
> We value the reviewer's insights, and we hope these clarifications address their concerns adequately.

---

### Official Review · Reviewer_UUNx · 2023-10-31

**Soundness:** 2 fair
**Presentation:** 2 fair
**Contribution:** 2 fair
**Rating:** 3
**Confidence:** 5

**Summary:**

This paper presents a video editing method with a video diffusion model. The proposed method can edit both motion and appearance, and it can acquire information from images or animate a single image. Dreamix achieves good results on the collected videos.

**Strengths:**

1. The proposed method can achieve various video editing and generation applications.
2. Extensive ablation studies are conducted to demonstrate the effectiveness of each component.

**Weaknesses:**

1. The comparison with previous works is not comprehensive. For text-based video editing, FateZero [ref-1] and TokenFlow [ref-2] are more advanced methods designed for video editing.
For subject-driven video generation, Animatediff [ref-3] can learn a Lora model from several images and generate corresponding videos. For animating a single image, VideoComposer [ref-4] can generate video conditioned on the single image and text, which does not require additional transformation. These methods should be discussed and compared in the experiments and related work.

[ref-1] Fatezero: Fusing attentions for zero-shot text-based video editing.

[ref-2] TokenFlow: Consistent Diffusion Features for Consistent Video Editing.

[ref-3] Animatediff: Animate your personalized text-to-image diffusion models without specific tuning.

[ref-4] VideoComposer: Compositional Video Synthesis with Motion Controllability.

2. The performance of motion editing is not satisfactory. This paper claims motion editing as one of the key contributions. However, as shown in Figure 7, the motion and appearance are from the video generation model (Uncond.), while Dreamix seems only to help to learn the background. From my perspective, motion editing should focus on keeping the same identity while changing the motion, instead of keeping the background.

3. The proposed method heavily relies on the choice of video generation model. For both single-image animation and subject-driven video generation, only frame-level finetuning is adopted ($\alpha$ in Eq.3 is 0). This requires the video generation model to handle single images, which may not be satisfied easily by other video generation models. It is better to verify the method on an open-source video generation model (e.g., [modelscope](https://huggingface.co/damo-vilab/modelscope-damo-text-to-video-synthesis))

4. During video editing, the degraded source video with noise is used as the initialization. In comparison, most video editing methods adopt DDIM inversion as initialization. It is better to compare the two methods.

5. **Minor**. I hope the authors can check the `\citep` and `\citet` in the LaTex template and improve the writing.

**Questions:**

See weaknesses.

---

> ### Author Response · Authors · 2023-11-17
>
> We appreciate the thorough review and have carefully considered their feedback:
>
> 1. **Baseline Comparisons:**
>     - **FateZero and TokenFlow:** We conducted preliminary experiments on these methods, and while they excel in appearance editing with impressive temporal consistency, they are unable to perform motion edits, which is the core of our work.
>
>     - **AnimateDiff:** We tested AnimateDiff on subject-driven video generation, but the results were not satisfactory. AnimateDiff, designed for small motion generation, faced challenges with the large motions in our examples (e.g., walking and drinking beer).
>
>     - **Image-driven video generation:** Following the reviewer's suggestion, we tested VideoComposer. In addition, we tested Gen-2, and PikaLabs (both commercial products). All methods failed to add significant motion or add objects to the input image and animate them.
>
> 2. **Motion Editing Capabilities:** We respectfully disagree with the reviewer's claim about Dreamix's motion editing. Dreamix learns both background and foreground, adding motion to both. While there might be cases where fidelity is slightly compromised for smoother videos with minor appearance changes, Dreamix is the only current method that is able to preserve such a high level of fidelity while still allowing for motion editing.  We refer the reviewer to our supplementary videos where the importance of Dreamix is more apparent.
>
> 3. **Applicability to Other VDM Backbones:** We conducted preliminary experiments, and the results align with our paper's claims that “Video diffusion models are general video editors”. Applying our method to the ModelScope backbone shows improved temporal consistency and editing capabilities compared to image-based methods but falls short of the power demonstrated by the much larger VDM backbone used in our paper.
>
> 4. **Initialization Method:** We agree with the reviewer that this is an interesting idea to investigate, however, this is out of scope for our current work.
>
> 5. **Citations:** We appreciate the suggestion and will review and improve the writing accordingly.
>
> We thank the reviewer for their insights, and we hope these clarifications address their concerns appropriately.

---

### Official Review · Reviewer_E6Fh · 2023-10-31

**Soundness:** 3 good
**Presentation:** 3 good
**Contribution:** 2 fair
**Rating:** 5
**Confidence:** 4

**Summary:**

This paper presents a novel extension to the video diffusion model by introducing video editing capabilities. The fine-tuning method of the combination of frame-level and video-level is used to ensure the motion and appearance quality. The experimental results demonstrate the model's effectiveness in video editing, image-to-video conversion and object-driven video generation.

**Strengths:**

1.	The model can accomplish multiple video editing tasks and show great visualized results.
2.	The fine-tuning strategy enhances the editing capabilities of VDM.
3.	The paper is well-structured, capable of clearly elucidating its core ideas.

**Weaknesses:**

1. This paper conducted editing experiments on a single VDM base model, making it difficult to ascertain whether the proposed method is applicable to other VDMs e.g., modelscope or if the observed results are solely due to the characteristics of the base model. This is somewhat inconsistent with the title of the paper.

2. The comparisons with Tune-A-Video are not fair. Tune-A-Video is finetuned on image diffusion model but dreamix is finetuned on video diffusion model.


3. When compared with other VDMs, only the motion editing capabilities (Figure 7) were assessed, and there is no comparison of appearance editing capabilities. This doesn't adequately showcase the impact of mixed-attention fine-tuning.

4.  Lacks specific information. Imagen Video operates on a series of diffusion models. It is unclear which components undergo fine-tuning during the process. Is there a necessity to fine-tune the super-resolution diffusion models as well? or only key frames diffusion model is enough.

**Questions:**

Please see the weakness.

---

> ### Author Response · Authors · 2023-11-17
>
> We appreciate the thoughtful review and aim to address their valuable feedback:
>
> 1. **Applicability to Other VDM Backbones:** Regarding the applicability to other video diffusion models such as ModelScope, we conducted preliminary experiments, and the results align with our paper's claims. Applying our method to the ModelScope backbone shows improved temporal consistency and editing capabilities compared to image-based methods but falls short of the power demonstrated by the much larger VDM backbone used in our paper.
>
> 2. **Tune-A-Video Comparisons:** We Acknowledge the concern about the comparison with Tune-A-Video, however, there are very few VDM-based editing methods. Following the reviewer's suggestion, we have also compared our method to the original version of ModelScope, the results reinforce our claims of improvement over the original ModelScope.
>
> 3. **Appearance Editing Comparisons:** Contrary to the reviewer's observation, we did assess both motion and appearance editing capabilities. Please refer to Tables 1 and 2, where the comparison results cover the entire dataset (detailed in Table 3).
>
> 4. **Implementation Details:** These details are available in Appendix C.2. In summary, our fine-tuning process involves the base model and the first two super-resolution models. For a comprehensive overview, please refer to the complete details in Appendix C.
>
> We hope these clarifications address the concerns raised by the reviewer, and we appreciate the opportunity to improve the paper based on their insightful feedback.

---

### Author Response · Authors · 2023-11-17

We express our sincere gratitude to all the reviewers for their thorough and insightful feedback on our paper. To address the concerns raised, we conducted extensive experiments, comparing our method with recent approaches and demonstrating its superiority. Additionally, based on the reviewers' valuable suggestions, we successfully applied our method to the publicly available ModelScope, showcasing the transferability of our results from the VDM backbone used in our paper to a public VDM. Despite these efforts, considering the overall scores received, we decided to withdraw the paper and submit it to another venue. We appreciate your time and thoughtful evaluation, which helped shape the future direction of our work.